# Amelioration of Obesity in Mice Fed a High-Fat Diet with Uronic Acid–Rich Polysaccharides Derived from *Tremella fuciformis*

**DOI:** 10.3390/polym14081514

**Published:** 2022-04-08

**Authors:** Chun-Hui Chiu, Kai-Chu Chiu, Li-Chan Yang

**Affiliations:** 1Graduate Institute of Health Industry and Technology, Research Center for Chinese Herbal Medicine and Research Center for Food and Cosmetic Safety, College of Human Ecology, Chang Gung University of Science and Technology, Taoyuan City 333, Taiwan; chchiu@mail.cgust.edu.tw; 2Department of Traditional Chinese Medicine, Chang Gung Memorial Hospital, Keelung 204, Taiwan; 3Master Program for Pharmaceutical Manufacture, China Medical University, Taichung 406, Taiwan; pto2398gq141@gmail.com; 4Department of Pharmacy, School of Pharmacy, China Medical University, Taichung 406, Taiwan

**Keywords:** high-fat diet, obesity, polysaccharide, *Tremella fuciformis*, uronic acid, viscosity

## Abstract

Obesity is rapidly becoming an emerging disease in developing countries due to the Westernization of societies and lifestyle changes. This study evaluated the ameliorative effect of acidic heteropolysaccharides derived from *Tremella fuciformis* (TFPS) on high-fat diet (HFD; 34.9% fat)-induced obesity in mice. The TFPS exhibited high uronic acid content and high viscosity in water. The structural characteristics of TFPS showed that average molecular weight was 679 kDa, and the monosaccharide composition was galactose, glucose, fructose, xylose, fucose, and mannose at a ratio of 1.0:6.5:10.0:18.5:30.5:67.5. In an in vivo study, HFD-induced obese C57BL/6 mice were orally given a TFPS treatment at 1 and 2 g/kg of body weight for 8 weeks. The TFPS treatment significantly reduced features of obesity in the mice, namely weight gain, feed efficiency, body fat percentage, and serum cholesterol level and increased fecal lipid content, compared with mice fed an HFD with water. In addition, TFPS exhibited the inhibition of cholesterol micelles in vitro in a concentration-dependent manner. In conclusion, the TFPS treatment ameliorated the diet-induced obesity in the mice, presumably reducing fat absorption in the intestine by interfering with viscous TFPS.

## 1. Introduction

Obesity is a condition in which the body is over deposited due to changes in physiological or biochemical functions, resulting in weight gain [1]. Obesity is rapidly becoming an emerging disease in developing countries due to the Westernization of societies and lifestyle changes [2,3]. Overweight and obesity are also world problems. According to the World Health Organization, approximately 1 billion people are overweight or obese [2]. Obesity, classified in terms of the body mass index and waist–hip ratio, has several associated comorbidities, namely diabetes mellitus, hypertension, degenerative osteoarthritis, and infertility [3]. Along with insufficient physical activity, the Westernization of diets has contributed to the global obesity epidemic. Fruits, vegetables, and whole grains are being replaced by readily accessible foods high in saturated fat, sugar, and refined carbohydrates [2]. An increased intake of dietary fiber appears to be effective in treating obesity [4]. Fiber-rich food is usually satisfying without being calorically dense [5]. Smith et al. [6] have reported that the supplementing a normal diet with gel-forming fibers increases satiation probably by slowing gastric emptying. Apart from a beneficial effect during caloric restriction, dietary fiber may improve some of the metabolic aberrations seen in obesity [5,6].

*Tremella fuciformis*, known as silver ear mushroom or white jelly fungus, is an edible basidiomycetous jelly mushroom that has been favored in Asia for centuries. The edible part of *T. fuciformis* is transparent to white with a jelly-like taste and is full of water soluble and indigestible polysaccharides. Polysaccharides are regarded as the major bioactive components of *T. fuciformis*, and studies have reported their bio-activities, such as humoral immune responses, antitumor, hypoglycernic, and hypolipidemic activities [7]. The structural characteristics of the polysaccharides derived from *T. fuciformis* contain glucuronoxylomannans, which consist of a linear (1 → 3)-linked-α-d-mannose backbone, mainly with β-d-xylose and β-d-glucuronic acid in the side chains [8,9]. A study [10] reported that these polysaccharides inhibited adipocyte differentiation through suppressing the expression of PPAR gamma, C/EBP alpha, and leptin; however, no study has investigated whether *T. fuciformis* polysaccharide supplements have anti-obesity effects in vivo. Therefore, this study examined the anti-obesity effects of the acidic heteropolysaccharides derived from *T. fuciformis* in a Western diet (high-fat diet) in mice.

## 2. Materials and Methods

### 2.1. Preparation of T. fuciformis Polysaccharides

The fruiting bodies of *T. fuciformis* were obtained from Well Youth Biotech Co., Ltd. (Taichung, Taiwan). Sawdust was used as the main cultivation substrate. The moisture content in the fresh fruiting bodies was 89–93%. To prepare the polysaccharides referred to by Yang et al., previously published [11], the fruiting bodies were cut into pieces and extracted by 10-fold of distilled water by weight at 80 °C for 4 h. Filtration was applied to remove debris, and 4-fold volume of 95% ethanol at 4 °C was added to the filtrate for 12 h to precipitate crude polysaccharides. A commercial total dietary fiber assay kit (K-TDFR, Megazyme, Ireland) was applied to the crude polysaccharides for removing the starch and proteins and obtained the indigestible *T. fuciformis* polysaccharide (TFPS) according to the AOAC 993.19 method. 

### 2.2. Characterization of Indigestible TFPS

The TFPS were purified and fractionated using anion-exchange chromatography on a DEAE-650M column (Tosho, Tokyo, Japan). The DEAE column was eluted with 20 mM of Tris–HCl (pH 7.8), followed by a sodium chloride gradient (0–0.33 M). The carbohydrate elution profiles of the TFPS were analyzed according to the phenol-sulfuric acid method, with glucose as the standard, and measured at 490 nm [12]. In a chemical analysis, the protein content of the elution was measured using a modified Bradford method assay with bovine serum albumin as the standard [13], whereas the carbohydrate content of the elution was determined using the phenol-sulfuric acid method [12]. The content of uronic acid was determined using the m-hydroxydiphenyl method, with glucuronic acid as the standard [14]. In a nuclear magnetic resonance (NMR) analysis, the major fraction from the DEAE elution (5 mg) was dissolved in deuterium oxide (0.5 mL), and ^1^H-NMR spectra were recorded on a Bruker DRX-600 spectrometer (Bruker BioSpin, Billerica, MA, USA) at 20 °C using 3-(trimethylsilyl)-propionic 2,2,3,3,-d4 acid sodium salt as an internal reference (δ 0.00 ppm). The relative molecular mass (Mr) was determined using high performance-size exclusion chromatography (HP-SEC), using a TSKgel PWHguard column with PWH (75 mm × 7.5 mm i.d.; TOSOH, Tokyo, Japan), TSKgel G4000PWXL, and G2500PWXL (300 mm × 7.8 mm i.d., TOSOH) columns connected in series. The columns were eluted with 0.3 N of a sodium nitrate solution containing 0.02% sodium azide at a flow rate of 0.8 mL/min at 65 °C. Peaks were detected using an interferometric refractometer (Wyatt Technology, CA, USA). The average Mr was estimated by comparison with the retention time of the pullulan standard P-82 kit (molecular weights of standards: P-800, 708 kDa; P-400, 375 kDa; P-200, 200 kDa; P-100, 107 kDa; P-50,47.1 kDa; P-20, 21.1 kDa; P-10, 9.6 kDa; and P-5, 5.9 kDa; Shodex, Kawasaki, Japan) and blue dextran 2000 (GE Healthcare, Uppsala, Sweden). The determination of molecular weights was also analyzed by dynamic light scattering (nanoPartica SZ-100V2, HORIBA, Kyoto, Japan). In a monosaccharide composition analysis, samples were hydrolyzed using a methanolysis method [15] for uronic acids analysis and trifluoroacetic acid (TFA) hydrolysis for neutral sugars. The resulting sample was analyzed using high-performance anion-exchange chromatography with pulsed amperometric detection (HPAEC-PAD). The peaks were detected using an 817 Bioscan PAD detector (Metrohm, Zofin-gen, Switzerland) and a CarboPac PA1 column (Dionex, Sunnyvale, CA, USA). The eluent consisted of 10 mM of sodium hydroxide and 1 mM of barium acetate (Sigma–Aldrich, MO, USA), pumped at a flow rate of 0.5 mL/min.

### 2.3. Determination of Viscosity and Inhibition on Cholesterol Micellization

Various concentrations of the TFPS were dissolved in distilled water (0–1.5%), and the viscosity was measured using a Sine-wave Vibro Viscometer SV-10 (A&D Co., Ltd., Tokyo, Japan) at a constant frequency of 30 Hz. The sample volume used in this experiment was 35 mL for each sample, measured at 25 °C, 37 °C, 50 °C, and 75 °C, respectively. As for the determination of inhibition on cholesterol micellization, a 15 mM phosphate buffered saline (PBS) (pH 7.4) containing 0.5 mM cholesterol, 4.8 mM lecithin, and 13.2 mM taurocholic acid was treated under ultrasonication at 37 °C for 24 h to prepare a cholesterol micellar solution [16]. In addition, 0.5 mL of TFPS solution in various concentrations (0–1.5%, *w*/*v*) was mixed with equal volume cholesterol micellar and incubated at 37 °C for 1 h. After incubation, the mixtures were centrifuged at 3000 rpm for 10 min and the supernatant was passed through a Millex-GP Syringe filter (0.22 μm, Millipore, Darmstadt, Germany), and the cholesterol in the filtrate was measured by a commercial cholesterol analysis kit (Fortress diagnostics, Antrim, UK). The inhibition of micellization was calculated by comparing the cholesterol level of the control group (TFPS 0 μg/mL) with the cholesterol level while the solution contained TFPS.

### 2.4. Animals and Experimental Design

The experimental animals received humane care, and the study protocols complied with the institutional guidelines of China Medical University for the use of laboratory animals (protocol No.: CMUIACUC-2017-286). The animals were housed in an air-conditioned room (21–24 °C) under 12 h of light (8:00 a.m.–8:00 p.m.) and were given free access to food pellets and water throughout the study. Eight-week-old male C57BL/6 mice were purchased from the National Laboratory Animal Center (Taipei, Taiwan). The mice were acclimated for 1 week and randomly divided into four groups (*n* = 12); normal diet (ND), high-fat diet (HFD) (control), HFD TFPS (low dose; HFD-TL), and HFD TFPS (high dose; HFD-TH). The ND mice were fed a maintenance diet (2.85 kcal/g, Altromin 1320, Altromin Spezialfutter GmbH & Co. KG, Im Seelenkamp, Germany), whereas the HFD mice were fed an HFD (D12492, Research Diet Inc., Brunswick, NJ, USA), which consisted of 34.9% fat and 5.24 kcal/g. The ND and HFD (control) mice were orally given distilled water (10 mL/kg body weight), and the HFD-TL and HFD-TH mice were given the TFPS at 1 and 2 g/kg of body weight, respectively, every day for 8 weeks using oral gavage. Body weight and food intake were measured twice a week, and unconsumed food was discarded. The formula for calculating calorie efficiency was as follows:kcal efficiency = (weight gain ÷ kcal intake) × 100%

The feed efficiency was calculated as follows:Feed efficiency = (weight gain ÷ food intake) × 100%

After 8 weeks of treatment, fresh feces were collected and stored at −20 °C until the fecal lipid content was analyzed. The mice were fasted overnight and sacrificed. Blood samples were collected and then centrifuged at 2000× *g* for 15 min for serum collection. Gonadal, retroperitoneal, and mesenteric fat were removed and weighed. The sum of the fat was the total body weight. The formula for calculating the body fat percentage was as follows:Body fat percentage = (total body fat ÷ body weight) × 100%

### 2.5. Serum Biochemistry Analysis

Levels of serum triacylglycerol (TG) were determined using a commercial kit (Roche, Indianapolis, IN, USA) and measured using an Automatic Biochemical Analyzer (COBAS MIRA PLUS, Roche, Indianapolis, IN, USA). Serum total cholesterol (T-Cho), high-density lipoprotein cholesterol (HDL-C), and low-density lipoprotein cholesterol (LDL-C) were determined using the peroxidase-antiperoxidase method and a commercial kit (BXC0262, BXC0442, BXC0432; Fortress Diagnostics Limited, UK). Non-esterified fatty acid (NEFA) content in the serum was determined using an assay kit (FA115, Randox, Antrim, UK), as per the manufacturer’s instructions.

### 2.6. Histology of the Gonadal Adipose Tissue

Gonadal adipose tissue was fixed with neutral 10% formaldehyde buffer and embedded in paraffin. Sections (5 μm) were stained with hematoxylin and eosin. The diameter of the adipose cells was scored and analyzed using Image-Pro (Plus 6, Media Cybernetics, Rockville, MD, USA).

### 2.7. Fecal Lipid Content Determination

After TFPS treatment for 8 weeks, all the mice were housed in individual metabolism cages containing a grid-floor and a facility for separate collection of feces for 3 days. Food consumption was also monitored on a daily basis when mice were in metabolism cages. The method for fecal lipid determination was as Kraus et al. [17] described before. Fecal samples of each animal were collected and weighted. Briefly, 1000 mg feces per mice were powderized using the tissue grinder, and then 5 mL of normal saline was added to 1000 mg of powderized feces in and vortexed. Next, 5 mL of Chloroform in methanol (2:1 by volume) was added to extract the lipids. The suspensions were centrifuged at 1000× *g* for 10 min at room temperature and then the chloroform/methanol phase was removed carefully to a weighted glass tube for evaporation under a fume hood. Then the tubes were weighted to obtain the lipid mass per 1000 mg of feces. To obtain the mouse’s average fecal lipid excretion per day, the lipid mass was multiplied by the total weight of feces in grams.

### 2.8. Statistical Analyses

The results were expressed as mean ± standard error. The statistical significance was evaluated using one-way analysis of variance (ANOVA), followed by Dunnett’s post hoc test using SPSS/11.5 software. Values were considered statistically significant at *p* < 0.05.

## 3. Results

### 3.1. Preparation and Characterization of the TFPS

The yield of the crude water-soluble polysaccharides was 60.1% ± 1.2% on a dry basis of *T. fuciformis*. TFPS was 95.2% ± 2.3% to the crude water-soluble polysaccharides. The TFPS consisted of 98.3% carbohydrates and less than 0.5% protein. The uronic acid content was 26.7% to TFPS on a carbohydrate basis.

The TFPS were fractioned using the DEAE-650M column to obtain the major fraction of the contained acidic polysaccharides (98%), which were collected according to the total carbohydrate elution profiles (Figure 1). The uronic acid content was determined from the elution; the results revealed that the TFPS were acidic polysaccharides rich in uranic acid (26.7% to total sugars). The ^1^H NMR (500 MHz) revealed the structural characteristics of the TFPS, as shown in Figure 2. A strong signal observed at 4.79 ppm corresponded to D_2_O, whereas signals at δ 5.12 and 5.49 ppm were consistent with the presence of α-d-annopyranose. A signal at δ 4.37 ppm potentially corresponded to β-d-xylopyranose. The signal for β-d-glucopyranuronic acid was not detected due to an overlap with the HOD peak [18]. Moreover, an obvious signal at δ 2.11 ppm corresponded to the CH_3_ moiety of the acetyl group [18]. Signals at δ 1.09~1.17 ppm potentially corresponded to the H2 of α-d-fucose, a deoxysugar. According to the calibration curves derived from the pullulan standards, the results of the HP-SEC analysis revealed that the average molecular weight of the TFPS was 679 kDa and dynamic light scattering analysis showed that the molecular weight of the TFPS was 736 kDa. The HPAEC-PAD analysis of the monosaccharide composition revealed that the TFPS were composed of galactose, glucose, fructose, xylose, fucose, mannose, and glucuronic acid at a ratio of 0.5: 3.6:5.5:10.1:16.7:36.9:26.7 (Appendix A).

### 3.2. Polysaccharide Viscosity and Effect on Cholesterol Micellization

The relationship between the TFPS concentration and viscosity in an aqueous solution is presented in Figure 3A. The viscosity of the TFPS solution increased as the concentration increased; additionally, the viscosity decreased in equivalent concentration of TFPS solution when the temperature increased. The viscosity of a 1.0% solution was approximately 88.3, 78.5, and 54.3 mPa.S at 25, 50, and 75 °C, respectively. The TFPS aqueous solution at a concentration of 1.8% became gelatinous without fluidity at 25 °C. In addition, the cholesterol micellization was inhibited while TFPS was added to cholesterol micelles solution in a dose-dependent manner (Figure 3B). When TFPS concentration increased, the date of viscosity concurred with the inhibition of cholesterol micellar.

### 3.3. Body Weight, Food Intake, and Tissue Weight

No differences were observed in the body weight of the mice among the groups at the start of the experiment. As shown in Figure 4, the body weight of the mice in the HFD groups was significantly higher than that of the mice in the ND group at day 7. Moreover, the body weight of the mice in the HFD-TL and HFD-TH groups was significantly lower than that of the mice in the control group from day 21 to the end of the experiment. No significant difference in body weight was observed between the HFD-TL and HFD-TH groups. Table 1 presents the food and calorie intake among the groups during the experiment. As shown, no significant differences were observed among the groups, even between the HFD and ND groups. Table 2 presents the weight gain and feed efficiency of the HFD and ND groups. The weight gain of the mice in the control group increased significantly compared with the mice in the ND group. The TFPS treatment significantly reduced the weight gain of the mice compared with the mice in the control group, which were also fed the HFD. No notable difference in total food intake was observed; however, the feed efficiency in the control group was 5.2-fold higher than that in the ND group. Moreover, the feed efficiency in the HFD-TL and HFD-TH groups was 0.71- and 0.65-fold higher than that in the control group, respectively.

At week 8, the weights of absolute adipose tissue were markedly greater in the HFD groups than in the ND group. Figure 5 shows that the weights of the gonadal, perirenal, and adipose tissues decreased significantly in both the TFPS-treated groups compared with the HFD-H_2_O group. The TFPS treatment exhibited effects in reducing the fat weight and body weight percentages (Figure 5).

### 3.4. Serum Lipid and Glucose Levels and Fecal Lipid Levels

No significant differences were found in the blood glucose levels among the groups at the end of the experiment. However, the blood sugar levels exhibited an increasing trend in the HFD groups compared with the ND group. Table 3 presents the serum lipid levels of the mice. The TG, T-Cho, LDL-C, and NEFA levels were significantly higher in the HFD-vehicle group compared with the ND group. The results revealed that the TFPS treatment reduced the T-Cho levels. Although the TFPS treatment did not significantly change the serum lipid levels, the HDL-C/LDL-C ratio was similar to that in the ND group. The HDL-C/LDL-C ratio was 1.75 and 1.26 in the ND and HFD-H_2_O groups, respectively. The results revealed that the HFD reduced the ratio in these groups. Moreover, the HDL-C/LDL-C ratio in the HFD-TL and HFD-TH groups was 1.44 and 1.47, respectively, significantly higher than that in the HFD-vehicle group. The results showed that fecal lipid in ND group was 8.9 ± 0.5 mg/day/mice and in HFD-H_2_O group was 11.2 ± 0.4 mg/day/mice. Fecal lipid content was 10.8 ± 0.6 and 9.1 ± 0.4 in the HFD-TL and HFD-TH group, respectively. A high dose of TFPS administration could significantly decrease the fecal lipid content compared to HFD-H_2_O group.

### 3.5. Histology of Gonadal Adipose Tissue

As shown in Figure 6, the HFD induced adipocyte hypertrophy in the gonadal adipose tissue compared with the ND-H_2_O group. The TFPS-treated groups exhibited less hypertrophy than the HFD-H_2_O group. The results from the other mice were similar to those presented in Figure 6.

## 4. Discussion

The species of *T. fuciformis* used in this study is called T8, which was selected from numerous species of *T. fuciformis* that have been collected in the mountains of Taiwan for several years. T8 has a larger fruiting body than other species and has natural fragrance smells like jasmine and lily floral. As one of the foremost medicinal and culinary mushrooms of China and Taiwan, *T. fuciformis* has the most unusual method of cultivation compared to other mushrooms, whereby it utilizes the nutrients from a common wood-decomposing ascomycete fungus, *Hypoxylon archeri*. The cultivation substrate used for T8 in this study was sawdust with rice bran as a supplementary nutrient, whereas cotton-seed hull is another common cultivation substrate used. 

The structural characterizations of the TFPS showed that mannose followed by fucose were the major monosaccharide constituents. Moreover, the TFPS were rich in glucuronic acid. The average molecular weight was 679 kDa. Another study [19] that investigated the polysaccharides in different phases of *T. fuciformis* reported that the polysaccharides from the mycelial phase had 13.3% (*w/w*) uronic acid. Yui et al. [8] reported that the heteropolysaccharide isolated from *T. fuciformis* has a linear backbone of 1,3-linked mannose with a highly beta-D-glucuronic acidic residue. The result of glyosidic linkage showed that the backbone of TFPS was mannan to glucuronic acid and fucose side chains are attached, and responded to the linkage types that Yui et al., reported [8]. In the current study, the monosaccharide composition was comparable to other reports [8,19]. The portion of fucose was relatively high in the TFPS and was similar to the portion of mannose. However, other studies [8,9,20] have reported that xylose was the second highest monosaccharide detected in the polysaccharides from *T. fuciformis.* The possible reasons for the difference in the monosaccharide composition among TFPS and other polysaccharides isolated *T. fuciformis* may be related to species, cultivation methods, or cultivation substrates. High fucose content seems to be one of the structural characteristics of TFPS derived from T8.

The average molecular weight (Mr.) of the TFPS was 679 kDa, which is similar to those of *T. fuciformis* polysaccharides presented in other studies. The Mrs of the aqueous extracts of *T. fuciformis* polysaccharides reported by Zhang et al. [21] and Liu et al. [18] were 599 and 582 kDa, respectively.

High molecular weight polymers greatly increase the viscosity of liquids in which they are dissolved. The intrinsic viscosity of a polymer relates to Mr, as per the Mark–Houwink rule, and MW distribution also affects the viscosity [22]. It is generally agreed that there is a positive relationship between intrinsic viscosity and degree of polymerization [23]. An increase in viscosity depends on the nature of both a polymer and solvent. The high portion of glucuronic acids in the TFPS caused a high charge density and water molecules to bond strongly, relative to the strength of the water–water interaction [24]. The results showed that the TFPS had a high molecular weight and high uronic acid/carbohydrate ratio, which might have caused their high viscosity. Moreover, the high viscosity of the TFPS in D_2_O potentially affected the ^1^H NMR signals from 3.4 to 4.5 ppm to be low in resolution (Figure 2). However, the HNMR spectra of the TFPS in this study were similar to those reported by Liu et al. [18].

The most recommended method for maintaining a healthy weight is to exercise and consume a healthy diet. Alternative weight loss treatments include medicines, surgery, and dieting. However, these treatments have side effects. Another alternative treatment for obesity is ingesting large amounts of dietary fiber. Numerous studies [25,26,27] have reported that an increase in dietary fiber can control body weight. Alginate, a matrix polysaccharide of brown algae, is a potential treatment for obesity [25,28]. Moreover, alginate has been added to beverages and cereal bars to increase satiety, reduce calorie intake, reduce fat digestion, and increase weight loss [25,29]. Oats, a cereal rich in β-glucans, have also exhibited obesity prevention and health benefits in metabolic syndrome [30,31]. Moreover, studies have shown that diets rich in fiber improve glycemic control in type 2 diabetes [32], reduce LDL-C in hypercholesterolemia [33,34], and contribute positively to long-term weight management [6,27]. The mechanism is believed to be due to the viscosity generated by β-glucans or other water-soluble dietary fibers, which influence gastrointestinal mechanisms that mediate satiety [27,35]. One of the possible mechanisms indicated that soluble polysaccharides could interfere with the absorption of cholesterol in the intestines [27]. Moreover, Pau-Roblot et al. [36] examined the relationship between polysaccharides uronic acid sequences and lipid molecules and found that uronic acids, like galacturonic and glucuronic acids in polysaccharides, have the ability to entrap cholesterol. This might explain why the TFPS in the present study, which were rich in uronic acid, affected the body lipid accumulation in the HFD mice. 

As reported [29,30,37], the polysaccharides derived from oats, plantago, and brown algae have the ability to reduce lipid uptake and hyperlipidemia in vivo. However, these polysaccharides do not have a similar backbone, and all of them have high viscosity. A study [6] reported that viscous fibers could be an adjunct to regular dietary treatment for obesity. Gel-forming fibers are particularly effective in reducing high LDL-cholesterol without changing the HDL-fraction [6]. Two structural characterizations of the TFPS, high uronic acid content and high viscosity, likely interfered with lipid absorption in the intestines of the mice on the HFD. In this study, TFPS exhibited the inhibition of cholesterol micelles in vitro and reduced the serum cholesterol level that was induced by the HFD in vivo. Both free or esterified cholesterol from diet should be emulsified by bile salts before absorption [38]. Various reports showed that polysaccharides with high viscosity and gel-forming property could influence the cholesterol absorption process. The possible mechanism may be that viscosity affects the diffusion of dietary mixed aggregates at the intestinal lumen, limiting cholesterol bio-accessibility. In addition, the interactions between polysaccharides and bile salts may reduce their emulsifying power towards cholesterol [39]. Heaton [40] has proposed that dietary fiber is a physiologic obstacle to energy intake by followed 3 mechanisms: (1) fiber replaces available calories and nutrients from the diet; (2) fiber increased satiety; and (3) fiber decreases the absorption efficiency of the small intestine. The third point is consistent with the assumption of this study that TFPS exhibits anti-obesity activity through interfering with lipid absorption in the intestine and results in increasing fecal lipid content. Moreover, the TFPS treatment increased the serum HDL-C/LDL-C ratio, which was similar to that in the normal group.

The results showed that the TFPS treatment reduced the weight gain in the HFD mice. Food and calorie intake in the two TFPS groups and HFD-vehicle group showed no significant difference. Moreover, the feed efficiencies of the two TFPS groups were significantly lower than that in the HFD-vehicle group. These results also revealed that the TFPS treatment did not affect appetite but influenced the lipid absorption in the HFD mice.

As mentioned, the daily calorie intake of the HFD-H_2_O group was not different from that of the ND group. Nevertheless, the body fat percentage, weight gain, and feed efficiency between those groups varied. The results confirmed that the different diets influenced the body fat accumulation of the mice. The TFPS treatment inhibited the HFD-induced weight gain, body fat accumulation, and serum lipid in vivo.

## 5. Conclusions

*T. fuciformis* contains a high amount of polysaccharides, which are rich in uronic acids and exhibit high viscosity when dissolved in water. The polysaccharides isolated from T8 have the potential to treat HFD-induced obesity. Administration of the TFPS treatment reduced body weight gain, body fat percentage, and serum total cholesterol in the HFD mice. The TFPS treatment also reduced hypertrophy in adipose tissue. Thus, the results suggest that the health benefits of TFPS in HFD-induced obesity may be associated with its influence on gut lipid absorption.

## Figures and Tables

**Figure 1 polymers-14-01514-f001:**
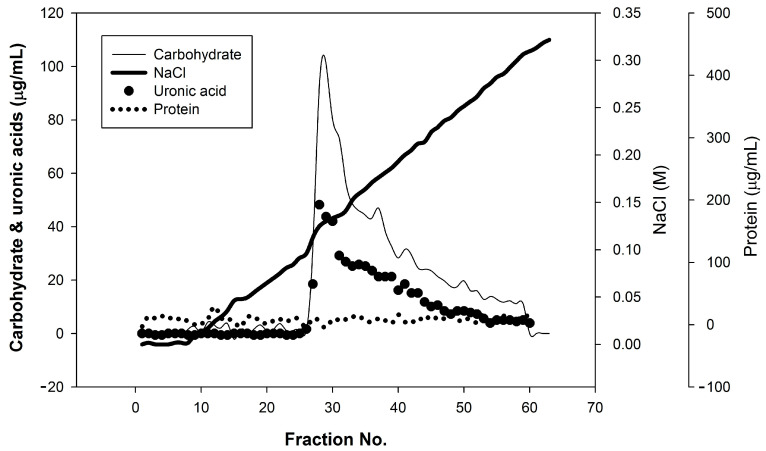
Anion-exchange chromatogram of water-soluble polysaccharides from *Tremella fuciformis* (TFPS) on a DEAE-650M column and eluted with 20 mM tris buffer (pH 7.8), followed by a sodium chloride gradient (0–0.32 M); the carbohydrate, uronic acid and protein contents were determined through a phenol–sulfuric acid method, *m*-hydroxydiphenyl method and Bradford method, respectively.

**Figure 2 polymers-14-01514-f002:**
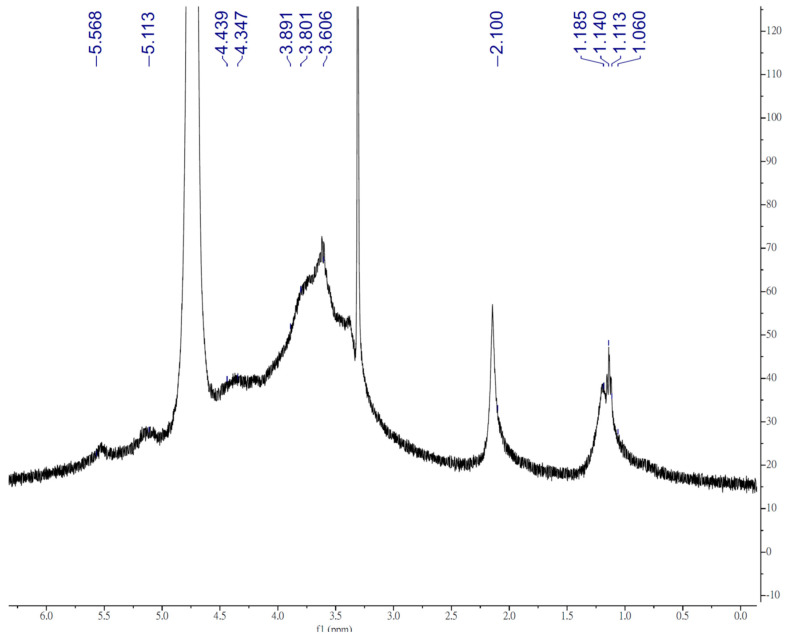
^1^H NMR spectra of polysaccharides from *Tremella fuciformis* (TFPS).

**Figure 3 polymers-14-01514-f003:**
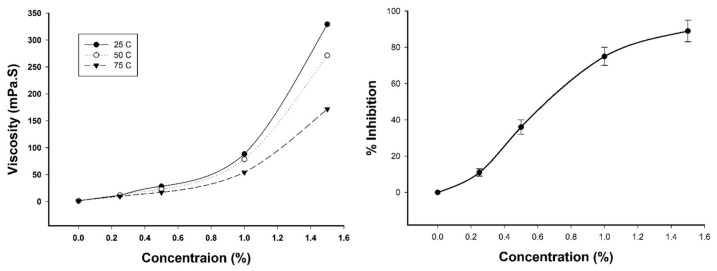
(**A**) Viscosity of aqueous solutions of polysaccharides from *Tremella fuciformis* (TFPS) in water at 25 °C, 37 °C, 50 °C, and 75 °C. (**B**) Effect of TFPS on the solubility of cholesterol micellar in vitro (*n* = 3).

**Figure 4 polymers-14-01514-f004:**
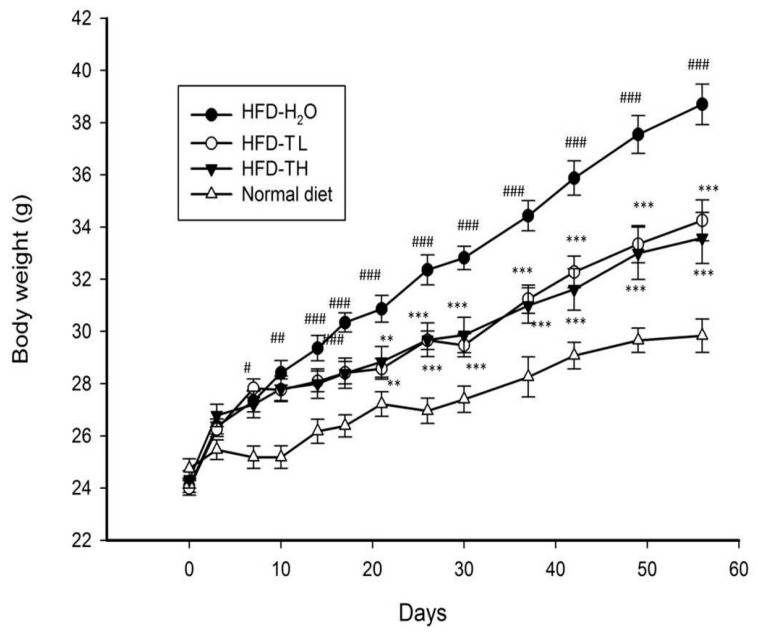
Body weight of mice in different groups. ^#^ *p* < 0.05, ^##^ *p* < 0.01, and ^###^ *p* < 0.001 while high-fat diet (HFD)-H_2_O group compared with the normal diet-H_2_O group. ** *p* < 0.01 and *** *p* < 0.001, while HFD-TFPS group compared with the HFD-H_2_O group. HFD-TL and HFD-TH groups received oral administration of 1 and 2 g/kg TFPS, respectively.

**Figure 5 polymers-14-01514-f005:**
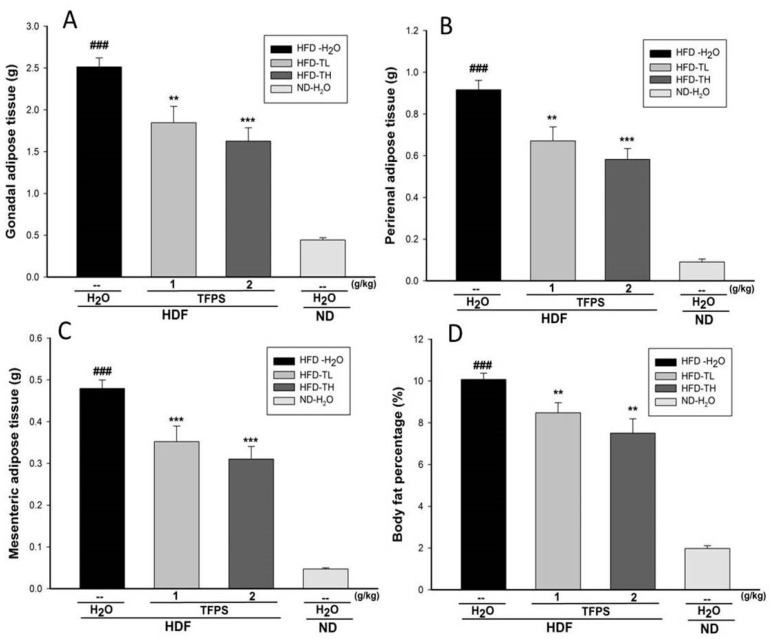
Effects of the TFPS treatment on the weights of the (**A**) gonadal adipose tissue, (**B**) perirenal adipose tissue, and (**C**) mesenteric adipose tissue as well as (**D**) body fat percentage. Mice were fed an HFD or ND, and the HFD mice were treated with water (*p.o.*) or TFPS (*p.o.*) for 8 weeks. All values are means ± standard error (*n* = 12). ^###^ *p* < 0.001 while HFD-H_2_O group compared with the ND-H_2_O group; ** *p* < 0.01 and *** *p* < 0.001 while HFD-TFPS groups compared with the HFD-H_2_O group.

**Figure 6 polymers-14-01514-f006:**
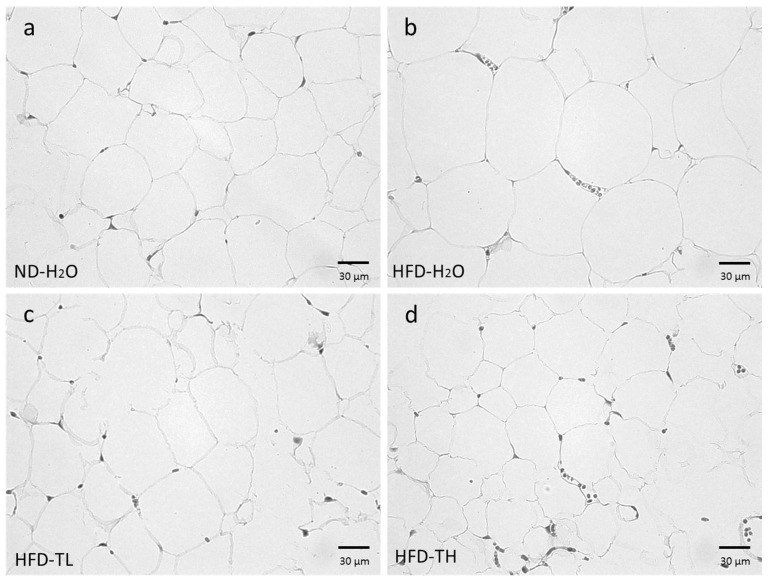
Histology of the gonadal adipose tissue of mice in the (**a**) ND-H2O, (**b**) HFD- H2O, (**c**) HFD-TL (TFPS 1 g/kg), and (**d**) HFD-TH (TFPS 2 g/kg) groups. Each image is representative of the nine mice (10 × 40).

**Table 1 polymers-14-01514-t001:** Food and calorie intake profiles at Week 8.

Weeks	Food Intake (g/day)			
	Normal Diet	High-Fat-Diet	High-Fat-Diet	High-Fat-Diet
	H_2_O	H_2_O	TFPS 1 g/kg	TFPS 2 g/kg
Week 1	4.1 ± 0.5	2.2 ± 0.1	2.4 ± 0.1	2.4 ± 0.2
Week 2	4.1 ± 0.2	2.3 ± 0.1	2.3 ± 0.1	2.2 ± 0.0
Week 3	4.5 ± 0.6	2.3 ± 0.1	2.2 ± 0.1	2.2 ± 0.1
Week 4	4.1 ± 0.4	2.3 ± 0.2	2.2 ± 0.1	2.2 ± 0.3
Week 5	4.1 ± 0.2	2.5 ± 0.1	2.3 ± 0.1	2.4 ± 0.3
Week 6	4.8 ± 0.8	2.5 ± 0.1	2.4 ± 0.2	2.3 ± 0.2
Week 7	3.2 ± 0.2	2.5 ± 0.2	2.4 ± 0.1	2.3 ± 0.1
Week 8	4.5 ± 0.2	2.5 ± 0.0	2.4 ± 0.1	2.3 ± 0.2
	Kcal intake (kcal/day)			
Week 1	11.8 ± 0.8	11.7 ± 0.2	12.4 ± 0.3	12.4 ± 0.6
Week 2	11.8 ± 0.3	12.0 ± 0.3	12.2 ± 0.4	11.8 ± 0.1
Week 3	12.9 ± 1.0	12.0 ± 0.3	11.6 ± 0.2	11.4 ± 0.3
Week 4	11.7 ± 0.6	12.3 ± 0.6	11.7 ± 0.4	11.6 ± 0.8
Week 5	11.8 ± 0.3	12.9 ± 0.3	12.3 ± 0.3	12.7 ± 1.0
Week 6	13.7 ± 1.3	13.2 ± 0.2	12.5 ± 0.5	12.0 ± 0.7
Week 7	12.2 ± 0.2	13.0 ± 0.6	12.7 ± 0.3	12.0 ± 0.2
Week 8	13.1 ± 0.3	13.1 ± 0.1	12.8 ± 0.4	12.1 ± 0.5

All data are presented as mean ± standard error (*n* = 12).

**Table 2 polymers-14-01514-t002:** Effects of the TFPS on weight gain and feed efficiency in mice.

Treatments		Dose (g/kg)	Weight Gain (g)	Feed Efficiency (%)
Normal diet	H_2_O	--	4.9 ± 0.4	2.1 ± 0.3
High-fat diet	H_2_O	--	14.6 ± 0.6 ^###^	11.0 ± 0.7 ^###^
	TFPS	1.0	10.3 ± 0.4 **	7.8 ± 0.2 ***
	TFPS	2.0	9.2 ± 0.5 ***	7.1 ± 0.2 ***

All data are presented as mean ± standard error (*n* = 12). ^###^ *p* < 0.001 high-fat diet (HFD)-H_2_O group compared with the normal diet-H_2_O group. ** *p* < 0.01 and *** *p* < 0.001, while HFD-TFPS groups compared with the HFD-H_2_O group.

**Table 3 polymers-14-01514-t003:** Serum lipid and glucose levels of mice in different groups.

Parameters	ND + H_2_O	HFD + H_2_O	HFD + TL	HFD + TH
TG (mg/dL)	92.4 ± 3.0	112.9 ± 4.2 ^###^	120.8 ± 5.0	122.1 ± 4.4
T-Cho (mg/dL)	61.1 ± 0.9	75.2 ± 2.0 ^###^	66.5 ± 3.0 *	64.8 ± 1.6 ***
LDL-C (mg/dL)	19.1 ± 1.6	30.9 ± 2.1 ^###^	26.6 ± 3.8	25.9 ± 3.4
HDL-C (mg/dL)	33.5 ± 3.5	38.9 ± 4.5	38.2 ± 5.8	38.2 ± 2.9
NEFA (mmol/L)	0.92 ± 0.23	1.40 ± 0.21 ^#^	1.25 ± 0.20	1.22 ± 0.13
Glc (mg/dL)	143.9 ± 10.2	159.1 ± 15.5	154.5 ± 19.7	160.0 ± 18.1

^###^ *p* < 0.001 HFD-H_2_O group compared with the ND-H_2_O group. * *p* < 0.05 and *** *p* < 0.001 HFD-TFPS groups compared with the HFD-H_2_O group. TFPS 1 g/kg and 2 g/kg was orally administered in the HFD-TL and HFD-TH groups, respectively.

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
