# Peer review of "Amelioration of Obesity in Mice Fed a High-Fat Diet with Uronic Acid–Rich Polysaccharides Derived from Tremella fuciformis"

_polymers, 2022, doi:10.3390/polym14081514_

Round 1

Reviewer 1 Report

This paper reports a polysaccharide from Tremella, Tremella fuciformis, which has the structural characteristics of glucuronox-ylomannans. Tremella fuciformis is rich in uronic acid, and the monosaccharide composition was galactose, glucose, fructose, xylose, fucose, and mannose. Tremella acid heteropolysaccharide has a certain improvement effect on high-fat diet-induced obesity in mice. In vivo studies found that oral administration of Tremella fuciformis to mice for eight weeks reduced features of obesity in the mice, which provides new ideas for related research. However, the points in the manuscript still need some refinement. The specific comments are as follows:

  1. Whether the food intake of each group of mice in Table 1 is artificially controlled, or whether the mice eat enough food by themselves. This should be explained.
  2. It is concluded that "The TFPS treatment exhibited a dose-dependent effect in reducing the fat weight and body weight percentages" in Figure 5. Whether the data support for the dose-dependent conclusion is incomplete, more groups of TFPS groups with different concentrations are needed.
  3. The mechanism of the concentration-dependent inhibitory effect of Tremella fuciformiss on cholesterol micelles should be properly explained.
  4. The mechanism of reducing obesity by TFPS should be given more discussion.
  5. The size and zeta potential of polysaccharide should be determined by DLS, the method can be according to recent report International Journal of Pharmaceutics 2022, 615, 121489.

Author Response

Dear reviewer,

Thanks for your valuable comments and suggestion. We have edited this MS carefully and the responses were as followed: 

  1. Whether the food intake of each group of mice in Table 1 is artificially controlled, or whether the mice eat enough food by themselves. This should be explained.

R:  In revised MS, page 6 line 146-147 showed that the animals were given free access to food pellets and water throughout the study.

  1. It is concluded that "The TFPS treatment exhibited a dose-dependent effect in reducing the fat weight and body weight percentages" in Figure 5. Whether the data support for the dose-dependent conclusion is incomplete, more groups of TFPS groups with different concentrations are needed.

R:   In revised MS, page 11, lines 267-268, the sentence was edited as followed “ The TFPS treatment exhibited a dose-dependent effects in reducing the fat weight and body weight percentages (Fig. 5)”

  1. The mechanism of the concentration-dependent inhibitory effect of Tremella fuciformiss on cholesterol micelles should be properly explained.

R:    Thanks for the valuable comment. The possible mechanism has be added into the discussion section in revised MS, page 15, lines 366-372, as followed “Both free or esterified cholesterol from dietary should be emulsified by bile salts before absorption [38]. Various reports showed that polysaccharides with high viscosity and gel-forming property could influence cholesterol absorption process. The possible mechanism may be that viscosity affect the diffusion of dietary mixed aggregates at the intestinal lumen, limiting cholesterol bio-accessibility. In addition, the interactions between polysaccharides and bile salts may reduce their emulsifying power towards cholesterol [39].”.

  1. The mechanism of reducing obesity by TFPS should be given more discussion.

R:    Thanks for the valuable comment. The possible mechanism has be added into the discussion section in revised MS, page 15, lines 372-378, as followed “ Heaton [40] has proposed that dietary fiber plays as a physiologic obstacle to energy intake by followed  3 mechanisms: (1) fiber replaces available calories and nutrients from the diet; (2) fiber increased satiety; and (3) fiber decreases the absorption efficiency of the small intestine. The third point is consistent to this study that TFPS was assumed to exhibit anti-obesity activity through interfering lipid absorption in intestine and result in increasing fecal lipid content.”

  1. The size and zeta potential of polysaccharide should be determined by DLS, the method can be according to recent report International Journal of Pharmaceutics 2022, 615, 121489.

 R:  Thanks for comment. We analyzed the Mr. of TFPS by DLS and get the Mr. was 736 kDa. The protocol and result of DLS was edited in the revised MS in page 5, lines 115-116 and page 10, lines 229-230.

Reviewer 2 Report

The article "Amelioration of obesity in mice fed a high-fat diet with uronic acid–rich polysaccharides derived from Tremella fuciformis" is very interesting and written well. Nice introduction is provided by reviewing the recent literature, however it can be improved by reviewing more recent literature regarding the use of polysaccharides for the treatment of obesity. Results were also presented in a good format, however there are few suggestions (please see below) to improve the article. Hence I recommend the article to publish after a major revision. 

1. Please expand the introduction by giving more information about how adipocyte inhibition linked to anti-obesity effects. 

2. Line 58-67: Please cite the reference article for the Preparation of T. fuciformis polysaccharides. 

3. Line 64: Expand AOAC.

4. Line 79: Correct the 1H-NMR spectra as "1H-NMR spectra".

5. Figure 2: 1H NMR spectrum is not in the good quality and the peaks are not clearly visible, and there is no signal visible at 4.37 ppm. It should be record again, I would suggest to record the 1H NMR for longer time (e.g.  more scans , up to 2-3 hours) to obtain good quality spectrum.

6. Please show the structure of your polysaccharides.

7. Line 188: please correct CH3 as "CH3". Which monosaccharide of your polysaccharide contain this acetyl group, and on which position?. Please also show it in the structure. 

8. Please present the HPAEC-PAD chromatogram.

9. Please separate the conclusion part from the discussion part by adding the sub title "conclusion".

10. "Author contributions" part is missing, please add it.

Author Response

Dear reviewer,

Thanks for your valuable comments and suggestion. We have edited this MS carefully and the responses were as followed: 

  1. Please expand the introduction by giving more information about how adipocyte inhibition linked to anti-obesity effects.

R:    The sentence for adipocyte inhibition was added in revised MS, page 4 line 72.

  1. Line 58-67: Please cite the reference article for the Preparation of T. fuciformis polysaccharides.

R:   Thanks for the comment and the reference was added, please see revised MS, page 4, line 82.

  1. Line 64: Expand AOAC.

R: the AOAC method was added in revised MS (page 4, lines 85-88).

  1. Line 79: Correct the 1H-NMR spectra as "1H-NMR spectra".

R: The words have been corrected, please see revised MS, page 5, line 102.

  1. Figure 2: 1H NMR spectrum is not in the good quality and the peaks are not clearly visible, and there is no signal visible at 4.37 ppm. It should be record again, I would suggest to record the 1H NMR for longer time (e.g. more scans , up to 2-3 hours) to obtain good quality spectrum.

R:    A new 1H NMR spectrum was replace to the original graph, please see revised MS (Fig. 2)

  1. Please show the structure of your polysaccharides.

R: The polysaccharide structure was very complex. For exhibition the structure, 2D NMR spectra with proper enzymatic digestion are required. Although current data in this MS could not for drawing the structure of TFPS and still provide enough information of polysaccharide structural characteristics to understand the features of TFPS.

  1. Line 188: please correct CH3 as "CH3". Which monosaccharide of your polysaccharide contain this acetyl group, and on which position?. Please also show it in the structure.

R: The word”CH3” has been corrected, please see page 9, line 225 in revised MS.

  1. Please present the HPAEC-PAD chromatogram.

R: The HPACE-PAD chromatogram was added in the supplement data in revised MS.

  1. Please separate the conclusion part from the discussion part by adding the sub title "conclusion".

R: Thanks for your comment. The section of Conclusion was separated.

  1. "Author contributions" part is missing, please add it.

R: Author contributions has added in revised MS as followed:

Author Contributions: Conceptualization, Yang‐L.C. and Chiu‐C.H.; Data curation, Yang‐L.C. and Chiu‐C.H; Formal analysis, Chiu‐C.H and Chiu‐K.C.; Methodology, Yang‐L.C. and Chiu‐K.C; Resources, Yang‐L.C. and Chiu‐C.H.; Supervision, Yang‐L.C.; Writing–original draft, Yang‐L.C. and Chiu‐C.H.; Writing–review & editing, Yang‐L.C. All authors have read and agreed to the published version of the manuscript.

Round 2

Reviewer 2 Report

The manuscript has been improved compared to the previous version and the authors have answered all my questions. I am satisfied with the authors' answers, so I recommend the article for publication in its current form.